# Embedding Inference
# for Structured Multilabel Prediction

**Farzaneh Mirzazadeh  Siamak Ravanbakhsh**
University of Alberta
{mirzazad,mravanba}@ualberta.ca

**Nan Ding**
Google
dingnan@google.com

**Dale Schuurmans**
University of Alberta
daes@ualberta.ca

## Abstract

A key bottleneck in structured output prediction is the need for *inference* during training and testing, usually requiring some form of dynamic programming. Rather than using approximate inference or tailoring a specialized inference method for a particular structure—standard responses to the scaling challenge—we propose to embed prediction constraints directly into the learned representation. By eliminating the need for explicit inference a more scalable approach to structured output prediction can be achieved, particularly at test time. We demonstrate the idea for multi-label prediction under subsumption and mutual exclusion constraints, where a relationship to maximum margin structured output prediction can be established. Experiments demonstrate that the benefits of structured output training can still be realized even after inference has been eliminated.

## 1  Introduction

Structured output prediction has been an important topic in machine learning. Many prediction problems involve complex structures, such as predicting parse trees for sentences [28], predicting sequence labellings for language and genomic data [1], or predicting multilabel taggings for documents and images [7, 8, 13, 20]. Initial breakthroughs in this area arose from tractable discriminative training methods—conditional random fields [19, 27] and structured large margin training [26, 29]—that compare complete output configurations against given target structures, rather than simply learning to predict each component in isolation. More recently, search based approaches that exploit sequential prediction methods have also proved effective for structured prediction [4, 21]. Despite these improvements, the need to conduct inference or search over complex outputs both during the training and testing phase proves to be a significant bottleneck in practice.

In this paper we investigate an alternative approach that eliminates the need for inference or search at test time. The idea is to shift the burden of coordinating predictions to the training phase, by embedding constraints in the learned representation that ensure prediction relationships are satisfied. The primary benefit of this approach is that prediction cost can be significantly reduced without sacrificing the desired coordination of structured output components.

We demonstrate the proposed approach for the problem of *multilabel classification* with hierarchical and mutual exclusion constraints on output labels [8]. Multilabel classification is an important subfield of structured output prediction where multiple labels must be assigned that respect semantic relationships such as subsumption, mutual exclusion or weak forms of correlation. The problem is of growing importance as larger tag sets are being used to annotate images and documents on the Web. Research in this area can be distinguished by whether the relationships between labels are assumed to be known beforehand or whether such relationships need to be inferred during training. In the latter case, many works have developed tailored training losses for multilabel prediction that penalize joint prediction behavior [6, 9, 30] without assuming any specific form of prior knowledge. More recently, several works have focused on coping with large label spaces by using low dimensional

projections to label subspaces [3, 17, 22]. Other work has focused on exploiting weak forms of prior knowledge expressed as similarity information between labels that can be obtained from auxiliary sources [11]. Unfortunately, none of these approaches strictly enforce prior logical relationships between label predictions. By contrast, other research has sought to exploit known prior relationships between labels. The most prominent such approaches have been to exploit generative or conditional graphical model structures over the label set [5, 16]. Unfortunately, the graphical model structures are either limited to junction trees with small treewidth [5] or require approximation [12]. Other work, using output kernels, has also been shown able to model complex relationships between labels [15] but is hampered by an intractable pre-image problem at test time.

In this paper, we focus on tractable methods and consider the scenario where a set of logical label relationships is given *a priori*; in particular, implication and mutual exclusion relationships. These relationships have been the subject of extensive work on multilabel prediction, where it is known that if the implication/subsumption relationships form a tree [25] or a directed acyclic graph [2, 8] then efficient dynamic programming algorithms can be developed for tractable inference during training and testing, while for general pairwise models [32] approximate inference is required. Our main contribution is to show how these relationships can be enforced without the need for dynamic programming. The idea is to embed label relationships as constraints on the underlying score model during training so that a trivial labelling algorithm can be employed at test time, a process that can be viewed as pre-compiling inference during the training phase.

The literature on multivariate prediction has considered many other topics not addressed by this paper, including learning from incomplete labellings, exploiting hierarchies and embeddings for multiclass prediction [31], exploiting multimodal data, deriving generalization bounds for structured and multilabel prediction problems, and investigating the consistency of multilabel losses.

## 2   Background

We consider a standard prediction model where a score function $s : \mathcal{X} \times \mathcal{Y} \to \mathbb{R}$ with parameters $\theta$ is used to determine the prediction for a given input $\mathbf{x}$ via

$$\hat{\mathbf{y}} \;=\; \arg\max_{\mathbf{y} \in \mathcal{Y}} s(\mathbf{x}, \mathbf{y}). \tag{1}$$

Here $\mathbf{y}$ is a *configuration* of assignments over a set of components (that might depend on $\mathbf{x}$). Since $\mathcal{Y}$ is a combinatorial set, (1) cannot usually be solved by enumeration; some structure required for efficient prediction. For example, $s$ might decompose as $s(\mathbf{x}, \mathbf{y}) = \sum_{c \in \mathcal{C}} s(\mathbf{x}, \mathbf{y}_c)$ over a set of cliques $\mathcal{C}$ that form a junction tree, where $\mathbf{y}_c$ denotes the portion of $\mathbf{y}$ covered by clique $c$. $\mathcal{Y}$ might also encode constraints to aid tractability, such as $\mathbf{y}$ forming a consistent matching in a bipartite graph, or a consistent parse tree [28]. The key practical requirement is that $s$ and $\mathcal{Y}$ allow an efficient solution to (1). The operation of maximizing or summing over all $\mathbf{y} \in \mathcal{Y}$ is referred to as *inference*, and usually involves a dynamic program tailored to the specific structure encoded by $s$ and $\mathcal{Y}$.

For supervised learning one attempts to infer a useful score function given a set of $t$ training pairs $(\mathbf{x}_1, \mathbf{y}_1), (\mathbf{x}_2, \mathbf{y}_2), ..., (\mathbf{x}_t, \mathbf{y}_t)$ that specify the correct output associated with each input. Conditional random fields [19] and structured large margin training (below with margin scaling) [28, 29] can both be expressed as optimizations over the score model parameters $\theta$ respectively:

$$\min_{\theta \in \Theta} \;\; r(\theta) + \sum_{i=1}^{t} \log \Big( \sum_{\mathbf{y} \in \mathcal{Y}} \exp(s_\theta(\mathbf{x}_i, \mathbf{y})) \Big) - s_\theta(\mathbf{x}_i, \mathbf{y}_i) \tag{2}$$

$$\min_{\theta \in \Theta} \;\; r(\theta) + \sum_{i=1}^{t} \max_{\mathbf{y} \in \mathcal{Y}} \Big( \Delta(\mathbf{y}, \mathbf{y}_i) + s_\theta(\mathbf{x}_i, \mathbf{y}) \Big) - s_\theta(\mathbf{x}_i, \mathbf{y}_i), \tag{3}$$

where $r(\theta)$ is a regularizer over $\theta \in \Theta$. Equations (1), (2) and (3) suggest that inference over $\mathbf{y} \in \mathcal{Y}$ is required at each stage of training and testing, however we show this is not necessarily the case.

**Multilabel Prediction**    To demonstrate how inference might be avoided, consider the special case of multilabel prediction with label constraints. Multilabel prediction specializes the previous set up by assuming $\mathbf{y}$ is a boolean assignment to a fixed set of variables, where $\mathbf{y} = (y_1, y_2, ..., y_\ell)$ and $y_i \in \{0, 1\}$, i.e. each label is assigned 1 (true) or 0 (false). As noted, an extensive literature that

has investigated various structural assumptions on the score function to enable tractable prediction. For simplicity we adopt the factored form that has been reconsidered in recent work [8, 11] (and originally [13]): $s(\mathbf{x}, \mathbf{y}) = \sum_k s(\mathbf{x}, y_k)$. This form allows (1) to be simplified to

$$\hat{\mathbf{y}} \quad = \quad \arg\max_{\mathbf{y} \in \mathcal{Y}} \sum_k s(\mathbf{x}, y_k) \quad = \quad \arg\max_{\mathbf{y} \in \mathcal{Y}} \sum_k y_k s_k(\mathbf{x}) \tag{4}$$

where $s_k(\mathbf{x}) := s(\mathbf{x}, y_k = 1) - s(\mathbf{x}, y_k = 0)$ gives the decision function associated with label $y_k \in \{0, 1\}$. That is, based on (4), if the constraints in $\mathcal{Y}$ were ignored, one would have the relationship $\hat{y}_k = 1 \Leftrightarrow s_k(\mathbf{x}) \geq 0$. The constraints in $\mathcal{Y}$ play an important role however: it has been shown in [8] that imposing prior implications and mutual exclusions as constraints in $\mathcal{Y}$ yields state of the art accuracy results for image tagging on the ILSVRC corpus. This result was achieved in [8] by developing a novel and rather sophisticated dynamic program that can efficiently solve (4) under these constraints. Here we show how such a dynamic program can be eliminated.

## 3 Embedding Label Constraints

Consider the two common forms of logical relationships between labels: implication and mutual exclusion. For *implication* one would like to enforce relationships of the form $y_1 \Rightarrow y_2$, meaning that whenever the label $y_1$ is set to 1 (true) then the label $y_2$ must also be set to 1 (true). For *mutual exclusion* one would like to enforce relationships of the form $\neg y_1 \vee \neg y_2$, meaning that at least one of the labels $y_1$ and $y_2$ must be set to 0 (false) (i.e., not both can be simultaneously true). These constraints arise naturally in multilabel classification, where label sets are increasingly large and embody semantic relationships between categories [2, 8, 32]. For example, images can be tagged with labels "dog", "cat" and "Siamese" where "Siamese" implies "cat", while "dog" and "cat" are mutually exclusive (but an image could depict neither). These implication and mutual exclusion constraints constitute the "HEX" constraints considered in [8].

Our goal is to express the logical relationships between label assignments as constraints on the score function that hold universally over all $\mathbf{x} \in \mathcal{X}$. In particular, using the decomposed representation (4), the desired label relationships correspond to the following constraints

$$\text{Implication} \quad y_1 \Rightarrow y_2: \quad s_1(\mathbf{x}) \geq -\delta \ \Rightarrow \ s_2(\mathbf{x}) \geq \delta \quad \forall \mathbf{x} \in \mathcal{X} \tag{5}$$
$$\text{Mutual exclusion} \ \neg y_1 \vee \neg y_2: \quad s_1(\mathbf{x}) < -\delta \ \text{ or } \ s_2(\mathbf{x}) < -\delta \quad \forall \mathbf{x} \in \mathcal{X} \tag{6}$$

where we have introduced the additional margin quantity $\delta \geq 0$ for subsequent large margin training.

### 3.1 Score Model

The first key consideration is representing the score function in a manner that allows the desired relationships to be expressed. Unfortunately, the standard linear form $s(\mathbf{x}, \mathbf{y}) = \langle \theta, f(\mathbf{x}, \mathbf{y}) \rangle$ cannot allow the needed constraints to be enforced over all $\mathbf{x} \in \mathcal{X}$ without further restricting the form of the feature representation $f$; a constraint we would like to avoid. More specifically, consider a standard set up where there is a mapping $f(\mathbf{x}, y_k)$ that produces a feature representation for an input-label pair $(\mathbf{x}, y_k)$. For clarity, we additionally make the standard assumption that the inputs and outputs each have independent feature representations [11], hence $f(\mathbf{x}, y_k) = \phi(\mathbf{x}) \otimes \psi_k$ for an input feature map $\phi$ and label feature representation $\psi_k$. In this case, a bi-linear score function has the form $s_k(\mathbf{x}) = \phi(\mathbf{x})^\top A \psi_k + b^\top \phi(\mathbf{x}) + c^\top \psi_k + d$ for parameters $\theta = (A, b, c, d)$. Unfortunately, such a score function does not allow $s_k(\mathbf{x}) \geq \delta$ (e.g., in Condition (5)) to be expressed over all $\mathbf{x} \in \mathcal{X}$ without either assuming $A = 0$ and $b = 0$, or special structure in $\phi$.

To overcome this restriction we consider a more general scoring model that extends the standard bi-linear form to a form that is linear in the parameters but *quadratic* in the feature representations:

$$-s_k(\mathbf{x}) \quad = \quad \begin{bmatrix} \phi(\mathbf{x}) \\ \psi_k \\ 1 \end{bmatrix}^\top \begin{bmatrix} P & A & \mathbf{b} \\ A^\top & Q & \mathbf{c} \\ \mathbf{b}^\top & \mathbf{c}^\top & r \end{bmatrix} \begin{bmatrix} \phi(\mathbf{x}) \\ \psi_k \\ 1 \end{bmatrix} \quad \text{for} \quad \theta \quad = \quad \begin{bmatrix} P & A & \mathbf{b} \\ A^\top & Q & \mathbf{c} \\ \mathbf{b}^\top & \mathbf{c}^\top & r \end{bmatrix}. \tag{7}$$

Here $\theta = \theta^\top$ and $s_k$ is linear in $\theta$ for each $k$. The benefit of a quadratic form in the features is that it allows constraints over $\mathbf{x} \in \mathcal{X}$ to be easily imposed on label scores via convex constraints on $\theta$.

**Lemma 1** *If $\theta \succeq 0$ then $-s_k(\mathbf{x}) = \|U\phi(\mathbf{x}) + \mathbf{u} - V\psi_k\|^2$ for some $U$, $V$ and $\mathbf{u}$.*

*Proof:* First expand (7), obtaining $-s_k(\mathbf{x}) = \phi(x)^\top P\phi(x) + 2\phi(x)^\top A\psi_k + 2\mathbf{b}^\top\phi(x) + \psi_k^\top Q\psi_k + 2\mathbf{c}^\top\psi_k + r$. Since $\theta \succeq 0$ there must exist $U$, $V$ and $\mathbf{u}$ such that $\theta = [U^\top, -V^\top, \mathbf{u}]^\top [U^\top, -V^\top, \mathbf{u}]$, where $U^\top U = P$, $U^\top V = -A$, $U^\top \mathbf{u} = \mathbf{b}$, $V^\top V = Q$, $V^\top \mathbf{u} = -\mathbf{c}$, and $\mathbf{u}^\top \mathbf{u} = r$. A simple substitution and rearrangement shows the claim. ∎

The representation (7) generalizes both standard bi-linear and distance-based models. The standard bi-linear model is achieved by $P = 0$ and $Q = 0$. By Lemma 1, the semidefinite assumption $\theta \succeq 0$ also yields a model that has a co-embedding [24] interpretation: the feature representations $\phi(\mathbf{x})$ and $\psi_k$ are both mapped (linearly) into a common Euclidean space where the score is determined by the squared distance between the embedded vectors (with an additional offset $\mathbf{u}$). To aid the presentation below we simplify this model a bit further. Set $\mathbf{b} = 0$ and observe that (7) reduces to

$$s_k(\mathbf{x}) = \gamma_k - \begin{bmatrix} \phi(\mathbf{x}) \\ \psi_k \end{bmatrix}^\top \begin{bmatrix} P & A \\ A^\top & Q \end{bmatrix} \begin{bmatrix} \phi(\mathbf{x}) \\ \psi_k \end{bmatrix} \qquad (8)$$

where $\gamma_k = -r - 2\mathbf{c}^\top\psi_k$. In particular, we modify the parameterization to $\theta = \{\gamma_k\}_{k=1}^\ell \cup \{\theta_{PAQ}\}$ such that $\theta_{PAQ}$ denotes the matrix of parameters in (8). Importantly, (8) remains linear in the new parameterization. Lemma 1 can then be modified accordingly for a similar convex constraint on $\theta$.

**Lemma 2** *If $\theta_{PAQ} \succeq 0$ then there exist $U$ and $V$ such that for all labels $k$ and $l$*

$$s_k(\mathbf{x}) = \gamma_k - \|U\phi(\mathbf{x}) - V\psi_k\|^2 \qquad (9)$$
$$\psi_k^\top Q\psi_k - \psi_k^\top Q\psi_l - \psi_l^\top Q\psi_k + \psi_l^\top Q\psi_l = \|V\psi_k - V\psi_l\|^2. \qquad (10)$$

*Proof:* Similar to Lemma 1, since $\theta_{PAQ} \succeq 0$, there exist $U$ and $V$ such that $\theta_{PAQ} = [U^\top, -V^\top]^\top [U^\top, -V^\top]$ where $U^\top U = P$, $V^\top V = Q$ and $U^\top V = -A$. Expanding (8) and substituting gives (9). For (10) note $\psi_k^\top Q\psi_k - \psi_k^\top Q\psi_l - \psi_l^\top Q\psi_k + \psi_l^\top Q\psi_l = (\psi_k - \psi_l)^\top Q(\psi_k - \psi_l)$. Expanding $Q$ gives $(\psi_k - \psi_l)^\top Q(\psi_k - \psi_l) = (\psi_k - \psi_l)^\top V^\top V(\psi_k - \psi_l) = \|V\psi_k - V\psi_l\|^2$. ∎

This representation now allows us to embed the desired label relationships as simple convex constraints on the score model parameters $\theta$.

## 3.2 Embedding Implication Constraints

**Theorem 3** *Assume the quadratic-linear score model (8) and $\theta_{PAQ} \succeq 0$. Then for any $\delta \geq 0$ and $\alpha > 0$, the implication constraint in (5) is implied for all $\mathbf{x} \in \mathcal{X}$ by:*

$$\gamma_1 + \delta + (1 + \alpha)(\psi_1^\top Q\psi_1 - \psi_1^\top Q\psi_2 - \psi_2^\top Q\psi_1 + \psi_2^\top Q\psi_2) \leq \gamma_2 - \delta \qquad (11)$$
$$\left(\tfrac{\alpha}{2}\right)^2 (\psi_1^\top Q\psi_1 - \psi_1^\top Q\psi_2 - \psi_2^\top Q\psi_1 + \psi_2^\top Q\psi_2) \geq \gamma_1 + \delta. \qquad (12)$$

*Proof:* First, since $\theta_{PAQ} \succeq 0$ we have the relationship (10), which implies that there must exist vectors $\nu_1 = V\psi_1$ and $\nu_2 = V\psi_2$ such that $\psi_1^\top Q\psi_1 - \psi_1^\top Q\psi_2 - \psi_2^\top Q\psi_1 + \psi_2^\top Q\psi_2 = \|\nu_1 - \nu_2\|^2$. Therefore, the constraints (11) and (12) can be equivalently re-expressed as

$$\gamma_1 + \delta + (1 + \alpha)\|\nu_1 - \nu_2\|^2 \leq \gamma_2 - \delta \qquad (13)$$
$$\left(\tfrac{\alpha}{2}\right)^2 \|\nu_1 - \nu_2\|^2 \geq \gamma_1 + \delta \qquad (14)$$

with respect to these vectors. Next let $\mu(\mathbf{x}) := U\phi(x)$ (which exists by (9)) and observe that

$$\begin{aligned}
\|\mu(\mathbf{x}) - \nu_2\|^2 &= \|\mu(\mathbf{x}) - \nu_1 + \nu_1 - \nu_2\|^2 \\
&= \|\mu(\mathbf{x}) - \nu_1\|^2 + \|\nu_1 - \nu_2\|^2 + 2\langle\mu(\mathbf{x}) - \nu_1, \nu_1 - \nu_2\rangle,
\end{aligned} \qquad (15)$$

Consider two cases.

*Case 1:* $2\langle\mu(\mathbf{x}) - \nu_1, \nu_1 - \nu_2\rangle > \alpha\|\nu_1 - \nu_2\|^2$. In this case, by the Cauchy Schwarz inequality we have $2\|\mu(\mathbf{x}) - \nu_1\|\|\nu_1 - \nu_2\| \geq 2\langle\mu(\mathbf{x}) - \nu_1, \nu_1 - \nu_2\rangle > \alpha\|\nu_1 - \nu_2\|^2$, which implies $\|\mu(\mathbf{x}) - \nu_1\| > \frac{\alpha}{2}\|\nu_1 - \nu_2\|$, hence $\|\mu(\mathbf{x}) - \nu_1\|^2 > \left(\frac{\alpha}{2}\right)^2 \|\nu_1 - \nu_2\|^2 \geq \gamma_1 + \delta$ by constraint (14). But this implies that $s_1(\mathbf{x}) < -\delta$ therefore it does not matter what value $s_2(\mathbf{x})$ has.

*Case 2:* $2\langle\mu(\mathbf{x}) - \nu_1, \nu_1 - \nu_2\rangle \leq \alpha\|\nu_1 - \nu_2\|^2$. In this case, assume that $s_1(\mathbf{x}) \geq -\delta$, i.e. $\|\mu(\mathbf{x}) - \nu_1\|^2 \leq \gamma_1 + \delta$, otherwise it does not matter what value $s_2(\mathbf{x})$ has. Then from (15) it follows that $\|\mu(\mathbf{x}) - \nu_2\|^2 \leq \|\mu(\mathbf{x}) - \nu_1\|^2 + (1 + \alpha)\|\nu_1 - \nu_2\|^2 \leq \gamma_1 + \delta + (1 + \alpha)\|\nu_1 - \nu_2\|^2 \leq \gamma_2 - \delta$ by constraint (13). But this implies that $s_2(\mathbf{x}) \geq \delta$, hence the implication is enforced. ∎

### 3.3 Embedding Mutual Exclusion Constraints

**Theorem 4** *Assume the quadratic-linear score model* (8) *and* $\theta_{PAQ} \succeq 0$. *Then for any* $\delta \geq 0$ *the mutual exclusion constraint in* (6) *is implied for all* $\mathbf{x} \in \mathcal{X}$ *by:*

$$\tfrac{1}{2}(\psi_1^\top Q\psi_1 - \psi_1^\top Q\psi_2 - \psi_2^\top Q\psi_1 + \psi_2^\top Q\psi_2) \;>\; \gamma_1 + \gamma_2 + 2\delta. \tag{16}$$

*Proof:* As before, since $\theta_{PAQ} \succeq 0$ we have the relationship (10), which implies that there must exist vectors $\nu_1 = V\psi_1$ and $\nu_2 = V\psi_2$ such that $\psi_1^\top Q\psi_1 - \psi_1^\top Q\psi_2 - \psi_2^\top Q\psi_1 + \psi_2^\top Q\psi_2 = \|\nu_1 - \nu_2\|^2$. Observe that the constraint (16) can then be equivalently expressed as

$$\tfrac{1}{2}\|\nu_1 - \nu_2\|^2 \;>\; \gamma_1 + \gamma_2 + 2\delta, \tag{17}$$

and observe that

$$
\begin{aligned}
\|\nu_1 - \nu_2\|^2 &= \|\nu_1 - \mu(\mathbf{x}) + \mu(\mathbf{x}) - \nu_2\|^2 \\
&= \|\nu_1 - \mu(\mathbf{x})\|^2 + \|\mu(\mathbf{x}) - \nu_2\|^2 + 2\langle\nu_1 - \mu(\mathbf{x}), \mu(\mathbf{x}) - \nu_2\rangle
\end{aligned} \tag{18}
$$

using $\mu(\mathbf{x}) := U\phi(x)$ as before (which exists by (9)). Therefore

$$
\begin{aligned}
\|\mu(\mathbf{x}) - \nu_1\|^2 + \|\mu(\mathbf{x}) - \nu_2\|^2 &= \|\nu_1 - \nu_2\|^2 - 2\langle\nu_1 - \mu(\mathbf{x}), \mu(\mathbf{x}) - \nu_2\rangle \\
&= \|(\nu_1 - \mu(\mathbf{x})) + (\mu(\mathbf{x}) - \nu_2)\|^2 - 2\langle\nu_1 - \mu(\mathbf{x}), \mu(\mathbf{x}) - \nu_2\rangle \quad (19) \\
&\geq \tfrac{1}{2}\|(\nu_1 - \mu(\mathbf{x})) + (\mu(\mathbf{x}) - \nu_2)\|^2 \quad (20) \\
&= \tfrac{1}{2}\|\nu_1 - \nu_2\|^2. \quad (21)
\end{aligned}
$$

(To prove the inequality (20) observe that, since $0 \leq \tfrac{1}{2}\|a - b\|^2$, we must have $\langle a, b\rangle \leq \tfrac{1}{2}\|a\|^2 + \tfrac{1}{2}\|b\|^2$, hence $2\langle a, b\rangle \leq \tfrac{1}{2}\|a\|^2 + \tfrac{1}{2}\|b\|^2 + \langle a, b\rangle = \tfrac{1}{2}\|a + b\|^2$, which establishes $-2\langle a, b\rangle \geq -\tfrac{1}{2}\|a + b\|^2$. The inequality (20) then follows simply by setting $a = \nu_1 - \mu(\mathbf{x})$ and $b = \mu(\mathbf{x}) - \nu_2$.)

Now combining (21) with the constraint (17) implies that $\|\mu(\mathbf{x}) - \nu_1\|^2 + \|\mu(\mathbf{x}) - \nu_2\|^2 \geq \tfrac{1}{2}\|\nu_1 - \nu_2\|^2 > \gamma_1 + \gamma_2 + 2\delta$, therefore one of $\|\mu(\mathbf{x}) - \nu_1\|^2 > \gamma_1 + \delta$ or $\|\mu(\mathbf{x}) - \nu_2\|^2 > \gamma_2 + \delta$ must hold, hence at least one of $s_1(\mathbf{x}) < -\delta$ or $s_2(\mathbf{x}) < -\delta$ must hold. Therefore, the mutual exclusion is enforced. ∎

Importantly, once $\theta_{PAQ} \succeq 0$ is imposed, the other constraints in Theorems 3 and 4 are all *linear* in the parameters $Q$ and $\boldsymbol{\gamma}$.

## 4 Properties

We now establish that the above constraints on the parameters in (8) achieve the desired properties. In particular, we show that given the constraints, inference can be removed both from the prediction problem (4) and from structured large margin training (3).

### 4.1 Prediction Equivalence

First note that the decision of whether a label $y_k$ is associated with $\mathbf{x}$ can be determined by

$$s(\mathbf{x}, y_k = 1) > s(\mathbf{x}, y_k = 0) \;\Leftrightarrow\; \max_{y_k \in \{0,1\}} y_k s_k(\mathbf{x}) > 0 \;\Leftrightarrow\; 1 = \arg\max_{y_k \in \{0,1\}} y_k s_k(\mathbf{x}). \tag{22}$$

Consider joint assignments $\mathbf{y} = (y_1, ..., y_l) \in \{0,1\}^l$ and let $\mathcal{Y}$ denote the set of joint assignments that are consistent with a set of implication and mutual exclusion constraints. (It is assumed the constraints are satisfiable; that is, $\mathcal{Y}$ is not the empty set.) Then the optimal joint assignment for a given $\mathbf{x}$ can be specified by $\arg\max_{\mathbf{y} \in \mathcal{Y}} \sum_{k=1}^{l} y_k s_k(\mathbf{x})$.

**Proposition 5** *If the constraint set* $\mathcal{Y}$ *imposes the constraints in* (5) *and* (6) *(and is nonempty), and the score function* $s$ *satisfies the corresponding constraints for some* $\delta > 0$, *then*

$$\max_{\mathbf{y} \in \mathcal{Y}} \sum_{k=1}^{l} y_k s_k(\mathbf{x}) = \sum_{k=1}^{l} \max_{y_k} y_k s_k(\mathbf{x}) \tag{23}$$

*Proof:* First observe that

$$\max_{\mathbf{y} \in \mathcal{Y}} \sum_{k=1}^{l} y_k s_k(\mathbf{x}) \leq \max_{\mathbf{y}} \sum_{k=1}^{l} y_k s_k(\mathbf{x}) = \sum_{k=1}^{l} \max_{y_k} y_k s_k(\mathbf{x}) \tag{24}$$

so making local classifications for each label gives an upper bound. However, if the score function satisfies the constraints, then the concatenation of the local label decisions $\mathbf{y} = (y_1, ..., y_l)$ must be jointly feasible; that is, $\mathbf{y} \in \mathcal{Y}$. In particular, for the implication $y_1 \Rightarrow y_2$ the score constraint (5) ensures that if $s_1(\mathbf{x}) > 0 \geq -\delta$ (implying $1 = \arg\max_{y_1} y_1 s_1(\mathbf{x})$) then it must follow that $s_2(\mathbf{x}) \geq \delta$, hence $s_2(\mathbf{x}) > 0$ (implying $1 = \arg\max_{y_2} y_2 s_2(\mathbf{x})$). Similarly, for the mutual exclusion $\neg y_1 \vee \neg y_2$ the score constraint (6) ensures $\min(s_1(\mathbf{x}), s_2(\mathbf{x})) < -\delta \leq 0$, hence if $s_1(\mathbf{x}) > 0 \geq -\delta$ (implying $1 = \arg\max_{y_1} y_1 s_1(\mathbf{x})$) then it must follow that $s_2(\mathbf{x}) < -\delta \leq 0$ (implying $0 = \arg\max_{y_2} y_2 s_2(\mathbf{x})$), and vice versa. Therefore, since the maximizer $\mathbf{y}$ of (24) is feasible, we actually have that the leftmost term in (24) is equal to the rightmost. ∎

Since the feasible set $\mathcal{Y}$ embodies non-trivial constraints over assignment vectors in (23), interchanging maximization with summation is not normally justified. However, Proposition 5 establishes that, if the score model also satisfies its respective constraints (e.g., as established in the previous section), then maximization and summation *can* be interchanged, and inference over predicted labellings can be replaced by greedy componentwise labelling, while preserving equivalence.

## 4.2 Re-expressing Large Margin Structured Output Training

Given a target joint assignment over labels $\mathbf{t} = (t_1, ..., t_l) \in \{0, 1\}^l$, and using the score model (8), the standard structured output large margin training loss (3) can then be written as

$$\sum_i \max_{\mathbf{y} \in \mathcal{Y}} \Delta(\mathbf{y}, \mathbf{t}_i) + \sum_{k=1}^{l} s(\mathbf{x}_i, y_k) - s(\mathbf{x}_i, t_{ik}) = \sum_i \max_{\mathbf{y} \in \mathcal{Y}} \Delta(\mathbf{y}, \mathbf{t}_i) + \sum_{k=1}^{l} (y_k - t_{ik}) s_k(\mathbf{x}_i), \tag{25}$$

using the simplified score function representation such that $t_{ik}$ denotes the $k$-th label of the $i$-th training example. If we furthermore make the standard assumption that $\Delta(\mathbf{y}, \mathbf{t}_i)$ decomposes as $\Delta(\mathbf{y}, \mathbf{t}_i) = \sum_{k=1}^{l} \delta_k(y_k, t_{ik})$, the loss can be simplified to

$$\sum_i \max_{\mathbf{y} \in \mathcal{Y}} \sum_{k=1}^{l} \delta_k(y_k, t_{ik}) + (y_k - t_{ik}) s_k(\mathbf{x}_i). \tag{26}$$

Note also that since $y_k \in \{0, 1\}$ and $t_{ik} \in \{0, 1\}$ the margin functions $\delta_k$ typically have the form $\delta_k(0, 0) = \delta_k(1, 1) = 0$ and $\delta_k(0, 1) = \delta_{k01}$ and $\delta_k(1, 0) = \delta_{k10}$ for constants $\delta_{k01}$ and $\delta_{k10}$, which for simplicity we will assume are equal, $\delta_{k01} = \delta_{k10} = \delta$ for all $k$ (although label specific margins might be possible). This is the same $\delta$ used in the constraints (5) and (6).

The difficulty in computing this loss is that it apparently requires an exponential search over $\mathbf{y}$. When this exponential search can be avoided, it is normally avoided by developing a dynamic program. Instead, we can now see that the search over $\mathbf{y}$ can be eliminated.

**Proposition 6** *If the score function* $s$ *satisfies the constraints in* (5) *and* (6) *for* $\delta > 0$, *then*

$$\sum_i \max_{\mathbf{y} \in \mathcal{Y}} \sum_{k=1}^{l} \delta(y_k, t_{ik}) + (y_k - t_{ik}) s_k(\mathbf{x}_i) = \sum_i \sum_{k=1}^{l} \max_{y_k} \delta(y_k, t_{ik}) + (y_k - t_{ik}) s_k(\mathbf{x}_i). \tag{27}$$

*Proof:* For a given $\mathbf{x}$ and $\mathbf{t} \in \mathcal{Y}$, let $f_k(y) = \delta(y, t_k) + (y - t_k)s_k(\mathbf{x})$, hence $y_k = \arg\max_{y \in \{0,1\}} f_k(y)$. It is easy to show that

$$1 \in \arg\max_{y \in \{0,1\}} f_k(y) \iff s_k(\mathbf{x}) \geq t_k\delta - (1 - t_k)\delta, \tag{28}$$

which can be verified by checking the two cases, $t_k = 0$ and $t_k = 1$. When $t_k = 0$ we have $f_k(0) = 0$ and $f_k(1) = \delta + s(\mathbf{x})$, therefore $1 = y_k \in \arg\max_{y \in \{0,1\}} f_k(y)$ iff $\delta + s(\mathbf{x}) \geq 0$. Similarly, when $t_k = 1$ we have $f_k(0) = \delta - s(\mathbf{x})$ and $f_k(1) = 0$, therefore $1 = y_k \in \arg\max_{y \in \{0,1\}} f_k(y)$ iff $\delta - s(\mathbf{x}) \leq 0$. Combining these two conditions yields (28).

Next, we verify that if the score constraints hold, then the logical constraints over $\mathbf{y}$ are automatically satisfied even by locally assigning $y_k$, which implies the optimal joint assignment is feasible, i.e. $\mathbf{y} \in \mathcal{Y}$, establishing the claim. In particular, for the implication $y_1 \Rightarrow y_2$, it is assumed that $t_1 \Rightarrow t_2$ in the target labeling and also that score constraints hold, ensuring $s_1(\mathbf{x}) \geq -\delta \Rightarrow s_2(\mathbf{x}) \geq \delta$. Consider the cases over possible assignments to $t_1$ and $t_2$:

If $t_1 = 0$ and $t_2 = 0$ then $y_1 = 1 \Rightarrow f_1(1) \geq f_1(0) \Rightarrow \delta + s_1(\mathbf{x}) \geq 0 \Rightarrow s_1(\mathbf{x}) \geq -\delta \Rightarrow s_2(\mathbf{x}) \geq \delta$ (by assumption) $\Rightarrow s_2(\mathbf{x}) \geq -\delta \Rightarrow \delta + s_2(\mathbf{x}) \geq 0 \Rightarrow f_2(1) \geq f_2(0) \Rightarrow y_2 = 1$.
If $t_1 = 0$ and $t_2 = 1$ then $y_1 = 1 \Rightarrow f_1(1) \geq f_1(0) \Rightarrow \delta + s_1(\mathbf{x}) \geq 0 \Rightarrow s_1(\mathbf{x}) \geq -\delta \Rightarrow s_2(\mathbf{x}) \geq \delta$ (by assumption) $\Rightarrow 0 \geq \delta - s_2(\mathbf{x}) \Rightarrow f_2(1) \geq f_2(0) \Rightarrow y_2 = 1$ (tight case).
The case $t_1 = 1$ and $t_2 = 0$ cannot happen by the assumption that $\mathbf{t} \in \mathcal{Y}$.
If $t_1 = 1$ and $t_2 = 1$ then $y_1 = 1 \Rightarrow f_1(1) \geq f_1(0) \Rightarrow 0 \geq \delta - s_1(\mathbf{x}) \Rightarrow s_1(\mathbf{x}) \geq -\delta \Rightarrow s_2(\mathbf{x}) \geq \delta$ (by assumption) $\Rightarrow 0 \geq \delta - s_2(\mathbf{x}) \Rightarrow f_2(1) \geq f_2(0) \Rightarrow y_2 = 1$.

Similarly, for the mutual exclusion $\neg y_1 \vee \neg y_2$, it is assumed that $\neg t_1 \vee \neg t_2$ in the target labeling and also that the score constraints hold, ensuring $\min(s_1(\mathbf{x}), s_2(\mathbf{x})) < -\delta$. Consider the cases over possible assignments to $t_1$ and $t_2$:

If $t_1 = 0$ and $t_2 = 0$ then $y_1 = 1$ and $y_2 = 1$ implies that $s_1(\mathbf{x}) \geq -\delta$ and $s_2(\mathbf{x}) \geq -\delta$, which contradicts the constraint that $\min(s_1(\mathbf{x}), s_2(\mathbf{x})) < -\delta$ (tight case).
If $t_1 = 0$ and $t_2 = 1$ then $y_1 = 1$ and $y_2 = 1$ implies that $s_1(\mathbf{x}) \geq -\delta$ and $s_2(\mathbf{x}) \geq \delta$, which contradicts the same constraint.
If $t_1 = 1$ and $t_2 = 0$ then $y_1 = 1$ and $y_2 = 1$ implies that $s_1(\mathbf{x}) \geq \delta$ and $s_2(\mathbf{x}) \geq -\delta$, which again contradicts the same constraint.
The case $t_1 = 1$ and $t_2 = 1$ cannot happen by the assumption that $\mathbf{t} \in \mathcal{Y}$.

Therefore, since the concatenation, $\mathbf{y}$, of the independent maximizers of (27) is feasible, i.e. $\mathbf{y} \in \mathcal{Y}$, we have that the rightmost term in (27) equals the leftmost. ∎

Similar to Section 4.1, Proposition 6 demonstrates that if the constraints (5) and (6) are satisfied by the score model $s$, then structured large margin training (3) reduces to independent labelwise training under the standard hinge loss, while preserving equivalence.

## 5 Efficient Implementation

Even though Section 3 achieves the primary goal of demonstrating how desired label relationships can be embedded as convex constraints on score model parameters, the linear-quadratic representation (8) unfortunately does not allow convenient scaling: the number of parameters in $\theta_{PAQ}$ (8) is $\binom{n+\ell}{2}$ (accounting for symmetry), which is quadratic in the number of features, $n$, in $\phi$ and the number of labels, $\ell$. Such a large optimization variable is not practical for most applications, where $n$ and $\ell$ can be quite large. The semidefinite constraint $\theta_{PAQ} \succeq 0$ can also be costly in practice. Therefore, to obtain scalable training we require some further refinement.

In our experiments below we obtained a scalable training procudure by exploiting trace norm regularization on $\theta_{PAQ}$ to reduce its rank. The key benefit of trace norm regularization is that efficient solution methods exist that work with a low rank factorization of the matrix variable while automatically ensuring positive semidefiniteness and still guaranteeing global optimality [10, 14]. Therefore, we conducted the main optimization in terms of a smaller matrix variable $B$ such that $BB^\top = \theta_{PAQ}$. Second, to cope with the constraints, we employed an augmented Lagrangian method that increasingly penalizes constraint violations, but otherwise allows simple unconstrained optimization. All optimizations for smooth problems were performed using LBFGS and nonsmooth problems were solved using a bundle method [23].

| Dataset | Features | Labels | Depth | # Training | # Testing | Reference |
|---------|----------|--------|-------|------------|-----------|-----------|
| Enron   | 1001     | 57     | 4     | 988        | 660       | [18]      |
| WIPO    | 74435    | 183    | 5     | 1352       | 358       | [25]      |
| Reuters | 47235    | 103    | 5     | 3000       | 3000      | [20]      |

Table 1: Data set properties

| % test error | Enron | WIPO | Reuters |
|--------------|-------|------|---------|
| unconstrained | 12.4 | 21.0 | 27.1 |
| constrained   | 9.8  | 2.6  | 4.0  |
| inference     | 6.8  | 2.7  | 29.3 |

| test time (s) | Enron | WIPO | Reuters |
|---------------|-------|------|---------|
| unconstrained | 0.054 | 0.070 | 0.60 |
| constrained   | 0.054 | 0.070 | 0.60 |
| inference     | 0.481 | 0.389 | 5.20 |

Table 2: (left) test set prediction error (percent); (right) test set prediction time (s)

# 6   Experimental Evaluation

To evaluate the proposed approach we conducted experiments on multilabel text classification data that has a natural hierarchy defined over the label set. In particular, we investigated three multi-label text classification data sets, Enron, WIPO and Reuters, obtained from `https://sites.google.com/site/hrsvmproject/datasets-hier` (see Table 1 for details). Some pre-processing was performed on the label relations to ensure consistency with our assumptions. In particular, all implications were added to each instance to ensure consistency with the hierarchy, while mutual exclusions were defined between siblings whenever this did not create a contradiction.

We conducted experiments to compare the effects of replacing inference with the constraints outlined in Section 3, using the score model (8). For comparison, we trained using the structured large margin formulation (3), and trained under a multilabel prediction loss without inference, but both including then excluding the constraints. For the multilabel training loss we used the smoothed calibrated separation ranking loss proposed in [24]. In each case, the regularization parameter was simply set to 1. For inference, we implemented the inference algorithm outlined in [8].

The results are given in Table 2, showing both the test set prediction error (using labelwise prediction error, i.e. Hamming loss) and the test prediction times. As expected, one can see benefits from incorporating known relationships between the labels when training a predictor. In each case, the addition of constraints leads to a significant improvement in test prediction error, versus training without any constraints or inference added. Training with inference (i.e., classical structured large margin training) still proves to be an effective training method overall, in one case improving the results over the constrained approach, but in two other cases falling behind. The key difference between the approach using constraints versus that using inference is in terms of the time it takes to produce predictions on test examples. Using inference to make test set predictions clearly takes significantly longer than applying labelwise predictions from either a constrained or unconstrained model, as shown in the right subtable of Table 2.

# 7   Conclusion

We have demonstrated a novel approach to structured multilabel prediction where inference is replaced with constraints on the score model. On multilabel text classification data, the proposed approach does appear to be able to achieve competitive generalization results, while reducing the time needed to make predictions at test time. In cases where logical relationships are known to hold between the labels, using either inference or imposing constraints on the score model appear to yield benefits over generic training approaches that ignore the prior knowledge. For future work we are investigating extensions of the proposed approach to more general structured output settings, by combining the method with search based prediction methods. Other interesting questions include exploiting learned label relations and coping with missing labels.

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
