[Reviews · NeurIPS 2015]

Submitted by Assigned_Reviewer_1

This paper presents an interesting method for dealing with certain restrictions on the space of label assignments in structured prediction settings by learning model potential functions with constraints that will enforce them.

This is in contrast with prevalent structured prediction methods that employ a search (i.e., dynamic programming) over the set of possible label assignments.

Specifically, implication and mutual exclusion constraints are considered and incorporated by constructing a quadratic expansion of the original feature function.

My questions are mainly on the expressivity of the model that results from these additional constraints.

If the computational concerns of Section 5 did not exist, should optimizing the larger set of parameters provide equivalent performance with the typical dynamic programming search structured model?

"Some preprocessing was performed on the label relations to ensure consistency with our assumptions" (line 377)

Please explain, this seems potentially worrisome depending on what modification of the data were required to match assumptions.

It seems that implication and mutual exclusion relationships could also be leveraged to drastically reduce the time complexity of dynamic programming if appropriately integrated. For example, ordering the inference from the "leaves" of implication trees.

Would this give comparable test time performance?

Or could you describe the degree of sophistication of the inference algorithm if it is already doing so?

Additionally, if I understand the label hierarchy setting correctly, it seems that the potentials from all ancestors in the label hierarchy could be summed and then the problem reduces to multi-class classification rather than a combinatorial search.

Is there other structure that prevents this?
Summary: This work tackles structured prediction with structured constraints on the label space by learning potentials that enforce these constraints rather than requiring a combinatorial search.

This is a novel and interesting idea, though I have some concerns about the experimental evaluations that should be addressed.

Submitted by Assigned_Reviewer_2

The paper proposes a specific approach to multi-label prediction with subsumption and mutual-exclusion constraints on categories.

First, a model is formulated as a quadratic parametrization, and then linear constraints are added which are proven to enforce the desired subsumption and mutual-exclusion constraints through the model without having to do expensive inference at test time.

Experimental results are shown on three text datasets.

Quality: The idea is a nice one, and multi-label inference with such constraints is argued to have given state of the art results in [10], which appears to be the main precursor to this submission.

The mathematical approach is not an obvious one, and seems to have nice computational properties.

The paper seems a bit rushed, with somewhat limited experimental results (one of the main numbers is missing, and apparently wasn't ready by the deadline), and no model selection performed for the regularization parameter.

This leaves a question open as to whether the additional constraints could lead to a slight difference in regularization between methods that can affect the results.

Also, would the difference in results lengthen (their method is faster, but less accurate than the competitor) with a different setting of the C parameter?

Clarity: The paper could be more clear in two main directions. (i) the presentation seems initially to focus on very general structured output prediction, but then the actual methods are relevant only for multi-label problems with a specific form of constraints.

(ii) the paper seems a bit rushed and many technical proofs that might otherwise be relegated to supplementary material are included in the main paper.

Originality: The idea is a nice one, and seems to advance [10] in a non-trivial way.

Significance: The approach effectively trades model expressiveness for efficiency, but in a way that seems to perform reasonably.

This is a reasonable thing to do in the accuracy/computation tradeoff curve.

line 286: "establishes is that"

========================================

Post rebuttal update: the rebuttal, additional experiments, and explanation that slowness was in part due to poorly chosen generic solver has convinced me to increase the score.

It seems that previous shortcomings were due to the paper being a bit rushed last minute, but there has been some improvement through the rebuttal.
Summary: The paper proposes a method for fast (trivial) inference in a specific structured-multi-label setting by enforcing well designed model constraints during learning.

This reduces the expressiveness of the model very slightly (shown in experimental results) for the benefit of much faster test time inference.

Submitted by Assigned_Reviewer_3

This paper presents an approach for learning multi-label prediction. Different from previous approaches that conduct inference to incorporate constraints between labels. The paper proposed to directly embed the implication and mutual exclusive constraints in the model. Although it seems to me the current approach is restricted to certain types of relations between labels, this direction is pretty interesting.

Quality and clarity: this paper is well-written and well-motivated. Experiment results support the claim of the paper.

Originality: There are lots of papers about multi-label classification, and I'm not familiar with many of them. However, to my knowledge, the approach introduced in the paper is new and interesting.

Significant: This paper proposed an interesting direction to deal with multi-label classification problems, although the current presentation seems restricted on certain types of multi-label problems (see below).

Comments/questions.

- I wonder if the proposed method is general and can apply to other constraints based on any first-order logic between labels.

- One of the main goal of the paper is to improve the speed in test time. However, I wonder if the proposed algorithm is efficient in the training time as well. I wonder if the authors can also show the training time performance of the proposed methods.
Summary: This paper presents an approach that directly embeds the constraints between labels in the model. This leads to a multi-label classifier that is very efficient in the test phase, while preserve a good test performance. Overall, the paper is interesting and well-written. Experiments on problems with many labels show the effectiveness of the proposed method.

Submitted by Assigned_Reviewer_4

This paper deals with a particular task of structured prediction: binary variables, score function consisting of unary potentials only, structure consists in the feasibility constraints (logical and mutual exclusion). The paper attempts to get rid of the costly inference at the test and train stages by posing special constraints on the model parameters such that trivial inference procedure automatically satisfies the feasibility constraints. The proposed approach is compatible with multiple structured losses, e.g. conditional log-likelihood (2) and structured large margin (3) are mentioned. To the best of my knowledge the idea is original and seems interesting. The paper is relatively clear although sometimes it is hard to follow the technical details (I have several major comments/questions about clarity/correctness, see below). The idea of embedding inference might be quite influential (lead to structured prediction methods without inference) even given that the direct applications of the paper are quite narrow.

Major: 1)

Results of sections 3.2 and 3.3 (key results of the paper) are based on Lemma 2, which has no proof. Lemma 2 consists of claims (9) and (10). I agree that (9) is analogous to lemma 1 and therefore clear. Equation (10), however, appears for the first time and in my opinion seems non-trivial. If it does not hold (under the assumptions of lemma 2) than section 3.2-3.3 are incorrect.

2) One of the contributions of the paper (mentioned in the abstract) is "equivalence to maximum margin structured output prediction". I do not see how this follows from the results of the paper. In particular, proposition 6 claims that is the score function satisfies constraints (5) and (6) than the large-margin objective can be computed without performing inference (r.h.s. of (22)). But general claim of equivalence in my opinion should sound something like this: "A minimizer of objective (3) (can be simplified by (22) under all the assumptions) under model (8) with theta_PAQ being positive semi-definite and constraints (11), (12), (16) (imposing (5) and (6)) at the same time minimizes objective (3) under model (8) with theta_PAQ being positive semi-definite (constraints are satisfied by inference)". I'd really like the authors to clarify what is meant by "equivalence". 3) In the experiments it is not specified how the hyper-parameters of the methods were chosen. In particular, choice of the weight of the regularizer r(theta) could significantly influence the results. Minor: Line 153. I do not understand why s_k(x) \geq 0 is a substantial requirement. Lines 159 and 319. Notation "argmax_{k \in {0,1} } f(k)= 1" is a bit confusing i.e. it is not clear what happens if f(1) = f(0). I suggest writing "1 \in argmax_{k \in {0,1} }". Line 295. Notation "t_{ik}" (two indices) is not introduced. Being slightly more careful would help the reader.

*Comments after the response Thanks for for prooving Lemma 2 and clarifying Proposition 6.

Please, incorporate these comments into the paper. In particular, phrase from the abstract "We demonstrate this idea for multi-label prediction under subsumption and mutual exclusion constraints, where equivalence to maximum margin structured output prediction is established." sounds like an overclaim.
Summary: This paper seems to be a solid contribution but I see some significant issues with clarity/correctness. If these issues are fixed I will vote for the acceptance.

Author Feedback
Author rebuttal: Thanks for the helpful comments!

Reviewer 1:

Regarding expressiveness: Given equivalent parameterizations (ignoring the computational concerns of Sec 5) the standard dynamic programming formulation can be seen as a form of relaxation of the constraint embedding approach: the standard approach does not require the learned parameterization to strictly enforce the constraints (instead it relies on the DP to enforce them post hoc), whereas the proposed approach requires the learned parameterization to enforce all constraints.

We explained the preprocessing method mentioned in Line 377 in Lines 392-3: we used the same assumptions as [10] and added constraints in the exact same manner as [10]. All competitors used the same data to ensure a fair comparison.

Unfortunately, it is not straightforward to leverage HEX constraints to reduce the time complexity of dynamic programming. The HEX graphs are usually not tree structured, since all siblings are mutually exclusive. In general, for exact inference, a junction tree needs to be constructed and belief propagation conducted over the cliques. In fact, exploiting the specific structure of HEX constraints to achieve an efficient dynamic program was the main result of [10]. [10] showed that the number of legal states in a HEX graph is usually linear to (as opposed to exponential in the general case), so the inference time could be significantly reduced. We have exploited all of the algorithmic techniques outlined in [10] to make our inference as efficient as possible in the comparison.

Generally a multi-label problem cannot be reduced to a multi-class problem, as suggested, without an exponential explosion in the number of classes. Even with the hierarchical constraint, the problem we consider is still multi-label at the leaves, for which there are still an exponential number of possible combinations.

Reviewer 2:

Unfortunately, the generic nonsmooth solver we were using [25] required quadratic space and was running out of memory (> 32G) for the standard linear representation used by the inference method on the Reuters data set. We have since replaced the solver and obtain a test error of 29.3% and test time of 5.30s for the n/a entries in the left and right tables in Table 2 respectively.

The results are quite robust to the choice of regularization parameter. For example, for the WIPO data set, the range of regularization parameter choices 0.01, 0.1, 1.0, 10.0 produced no change in the test accuracies.

Reviewer 3:

1) Indeed (10) holds under the assumptions of Lemma 2.
Proof: If $\theta_{PAQ} \succeq 0$, then we will have that $Q \succeq 0$ (any principal submatrix of a PSD matrix, is PSD). So we can write Q as $Q = V^TV$, for some $V$. Then the LHS of (10) can be equivalently written as (10) = $(\psi_k - \psi_l)^T Q (\psi_k - \psi_l) = (\psi_k - \psi_l)^T V^T V (\psi_k - \psi_l) = \|V\psi_k - V\psi_l \|^2$.

2) Proposition 6 established that for any score model satisfying (5) and (6), regardless of parameterization, the objective value obtained with inference (LHS of (27)) is equal to the decomposed objective value obtained without inference (RHS of (27)). Therefore, to be precise, the notion of equivalence is a conditional one: if the constraints (5) and (6) are satisfied, then minimizing the standard structured output loss (LHS of (27)) is equivalent to minimizing the decomposed loss (RHS of (27)). But, in general, a minimizer of the LHS of (27) need not satisfy the constraints (5) and (6).

3) In the submission, we merely set the regularization parameter to 1.0. The results are quite robust to this choice (above).

Line 153: To ensure, say, $y_i \Rightarrow y_k$ in (5), one needs to enforce $s_k(x) \ge \delta, \forall x$; unfortunately enforcing $s_k(x) \ge 0$ is provably impossible with a non-vacuous bilinear score model.

Lines 159 (259?) and 319: Thanks

Line 295: Thanks, $t_{ik}$ denotes the k-th true label of the i-th training example.

Reviewer 4:

(5) has no typographical error: The condition in (5) asserts that if $s_1$ is close to (but still below) zero by a margin, then $s_2$ must be above zero (by a margin). This condition is needed to establish the relationship to standard large margin training given in Section 4.

Reviewer 5:

The logics we have consider are generally weaker than first order logic. Using (8) under the theta_PAQ is positive semi-definite constraint, we can show that constraints at least as logically expressive as the diagrammatic language of $d$-dimensional Euler diagrams can be expressed; see Section 1.1.1 of: Gem Stapleton, "Reasoning with constraint diagrams", PhD Dissertation, University of Brighton, 2004.

Reviewer 6:

We have used the HEX constraints proposed in [10] as a clear and concrete demonstration of the idea. More general types of constraints can easily be expressed, such as Horn clauses, and constraints expressible by Euler diagrams (as noted above).